# Criteria to define rare diseases and orphan drugs: a systematic review protocol

Ghada Mohammed Abozaid [1,2] Katie Kerr [2] Amy McKnight [2]
Hussain A Al-Omar [3,4,5]

¹Department of Pharmacy Practice, Princess Nourah bint Abdulrahman University, Riyadh, Saudi Arabia
²Institute of Clinical Sciences B, Royal Victoria Hospital, Queen's University Belfast School of Medicine, Dentistry and Biomedical Sciences,Centre for Public Health, Belfast, UK
³Deparment of Clinical Pharmacy, College of Pharmacy, King Saud University, Riyadh, Saudi Arabia
⁴Center of Health Technology Assessment, Ministry of Health, Riyadh, Saudi Arabia
⁵Health Technology Assessment Unit (HTAU), College of Pharmacy, King Saud University, Riyadh, Saudi Arabia

**Correspondence to**
Hussain A Al-Omar;
halomar@ksu.edu.sa

## ABSTRACT

**Introduction** Rare diseases (RDs) are often chronic and progressive life-threatening medical conditions that affect a low percentage of the population compared with other diseases. These conditions can be treated with medications known as orphan drugs (ODs). Unfortunately, there is no universal definition of RDs or ODs. This systematic review (SR) will identify the quantitative and qualitative criteria and the underlying rationale used internationally to define RDs and ODs.

**Methods and analysis** This protocol follows the conventions for the Preferred Reporting Items for Systematic Review and Meta-Analysis Protocols (2015 guidelines). A SR will be conducted, including a search of the following databases: PubMed, MEDLINE, EMBASE, Scopus, Web of Science, GreyLit and OpenGrey. Eligible publications will be selected based on predetermined inclusion criteria. Extracted data will be analysed using thematic and content analyses of qualitative descriptors, whereas quantitative data will be analysed descriptively and reported in the form of frequencies and percentages.

**Ethics and dissemination** No ethical approval is required since this SR focuses on the secondary analysis of data retrieved from the scientific literature. The outcomes of this SR will be published as part of a PhD thesis, presented at conferences, and published in peer-reviewed journals.

**PROSPERO registration number** CRD42021252701.

## STRENGTHS AND LIMITATIONS OF THIS STUDY

⇒ This document describes a comprehensive systematic review following the Preferred Reporting Items for Systematic Reviews and Meta-Analysis Protocols (2015 guidelines).
⇒ It summarises the definitions and criteria retrieved from the literature in relation to study design, geographical location, methodological rigour and outcomes.
⇒ Screening of the articles, data extraction, quality and risk of bias assessment will be conducted independently by two reviewers, with an independent third reviewer resolving any disagreements.
⇒ The quality of evidence for all outcomes will be judged using the Joanna Briggs Institute's critical appraisal tools for different articles and an authority, accuracy, coverage, objectivity, date and significance checklist for grey literature.
⇒ Relevant studies published in non-English languages may be missed by this review.

## INTRODUCTION

A rare disease (RD) is a health condition with a low prevalence compared with common diseases.[1] RDs affect approximately 6% of the worldwide population,[2] when extrapolated to the worldwide population in 2022 (7 950 418 550)[3] that is equivalent to >470 million people. Many patients with RDs experience difficulties accessing appropriate treatment options. Globally, less than one-tenth of patients with RDs receive disease-specific treatment.[4] The varied terminology and inconsistent definitions of RDs are considered major challenges in treatment accessibility. The insufficient data relating to what diseases are considered rare deliberated as an obstacle in understanding of these diseases, defining them, disease coding, correctly diagnosed and gain pharmaceutical interest. Unfortunately, there is no single, unified and universally accepted definition of RD.[5]

Several international definitions of RD have been proposed and used based on different stakeholders' priorities and perspectives.[5] Stakeholders include scientific societies, patient groups, regulatory agencies, industry, reimbursement agencies, payers, decision-makers and policy-makers. For instance, payers view RDs from healthcare spending and resource utilisation standpoints, which is different from the perspectives of patient groups because their primary goal is often focused on treatment accessibility, as opposed to that of policy-makers, whose priority is improving the efficiency of healthcare delivery and the healthcare system. Therefore, it is important to understand the context and application of the definitions for RD used by different stakeholders.[5]

The criteria used to define RD also differ between countries and organisations. These

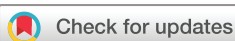

may be based on either qualitative (subjective) or quantitative (objective and measurable) descriptors. Some definitions use qualitative norms related to emotional connections, such as disease severity, whether it is life threatening or inheritable, or the availability of alternative treatments.[5] [6] These subjective descriptors are not supported by substantive evidence; they differ depending on the type of organisation using the term. For example, the European Organisation for Rare Diseases incorporates both RDs and neglected diseases under the category of 'orphan diseases'.[7] Quantitative descriptors, such as the disease prevalence threshold, are the preferred epidemiology-measured elements used in 58% of RD definitions, either explicitly or implicitly based on data from 2014.[5] Prevalence describes the data on the basis of the proportion of a particular population living with a disease at a specific time,[8] while incidence reflects the rate of occurrence. Prevalence data can help determine the needs of patients with RD at a population level when conducting economic evaluations, estimating the financial resources required for health and social services, estimating disease burdens and offering optimal designs for clinical trials.[2] However, determining a prevalence threshold is challenging as it originated from a variety of different information sources. This is heightened by lack of determinedly diagnostic criteria or coding systems to obtain this data.[9] Consequently, some diseases might be considered rare in one country but not in another, due to genetic population diversity, environmental or societal pressures, or survival issues in different regions.[10]

In the United Kingdom (UK), the 2021 Rare Disease Framework defined RD based on prevalence, as a condition affecting fewer than 1 in 2000 people.[11] Meanwhile, European Union (EU) countries use qualitative and quantitative criteria to define RDs as being 'life-threatening or chronically debilitating diseases which are of such low prevalence (less than 5 per 10 000) that special combined efforts are needed to address them so as to prevent significant morbidity or perinatal or early mortality or a considerable reduction in an individual's quality of life or socioeconomic potential'.[12] In the United States of America (USA), the Rare Diseases Act (RDA) of 2002,[13] precisely defines RD according to prevalence: "rare disease' means any disease or condition that affects less than 200 000 persons in the USA'. Prior to the RDA, the Orphan Drug Act (ODA) of 1983[14] was the federal law, which facilitated the development of RD drugs, and it defined RDs based on qualitative descriptors as follows: 'the term 'rare disease or condition' means any disease or condition which occurs so infrequently in the USA that there is no reasonable expectation that the cost of developing and making available in the USA a drug for such disease or condition will be recovered from sales in the USA of such drug'. Meanwhile, the Food and Drug Administration (FDA) used both qualitative and quantitative criterion to define RD as 'any disease or condition that affects less than 200 000 people in the USA or affects >200 000 in the USA and for which there is no

reasonable expectation that the cost of developing and making available in the USA a drug for such disease or condition will be recovered from sales in the USA of such drug'.[15] [16] The criteria used in Australia and Japan assume that an RD affects 11 or 40 out of 100 000 people, respectively.[17] Although these definitions are widely used, some countries have adopted others. Argentina, for instance, identifies RD using the EU definition,[18] while Brazil fosters the World Health Organisation definition of 'those affecting less than 65 out of 100 000 individuals'.[19]

As RDs each individually affect a small number of people, they are 'orphaned' by the pharmaceutical industry, which has promoted the use of the term 'orphan disease' when referring to rare conditions.[20] To address the therapeutic gap in treatment options for orphan diseases, several countries have established mechanisms to encourage pharmaceutical companies to invest in, and manufacture, orphan drugs (ODs).[4 17 18 21]

Being termed an OD is a designation status issued by regulatory bodies to describe therapies that treat rare conditions.[22] [23] Pharmaceutical companies that apply for such a designation are incentivised by these regulatory bodies if their therapy fulfils the designation criteria.[22] [23] OD definitions vary beyond national boundaries. The UK,[24] for instance, shares the same criterion as the European Medicines Agency, which is 'if the drug is intended for the diagnosis, prevention or treatment of a life-threatening or chronically and seriously debilitating condition affecting not more than 5 in 10 000 EU people or that it is unlikely that marketing the drug in the EU would generate sufficient benefit for the affected people and for the drug manufacturer to justify the investment'.[21] Meanwhile, the FDA defines an OD as 'one intended for the treatment, prevention or diagnosis of a rare disease or condition, which is one that affects less than 200 000 persons in the USA' (which equates to approximately 6 cases per 10 000 population) 'or meets cost recovery provisions of the act'.[25 26] Both definitions depend on the disease prevalence as a criterion for OD designation. In contrast, other countries may have more than one criterion, including prevalence, for designating an OD. For instance, in Japan, to designate a drug as an orphan, the drug must meet these three criteria: the drug is used to treat an RD, there are no other treatments available in Japan or the proposed drug is clinically superior to drugs already available on the Japanese market.[21] According to Section 16H of the Therapeutic Goods Regulations 1997 in Australia, ODs are defined as 'a medicine, vaccine or in vivo diagnostic agent consider an orphan drug if it intended to treat, prevent or diagnose a rare disease or must not be commercially viable to supply to treat, prevent or diagnose another disease or condition'.[21 27]

As already mentioned, some countries foster an international RD definition by using other countries' criteria, which can be problematic due to differences in regional demographic, governmental, economic and socio-cultural considerations.[5] For instance, consanguineous marriage (CM) is a major risk factor for congenital

abnormalities and genetic diseases that are inherited in an autosomal recessive manner, resulting in some RDs.[28] In Saudi Arabia, there is no national definition for RD or OD, although CM represents 70% of the total marriages in the country.[28–31] Therefore, it could be considered improper for Saudi Arabia to espouse or adopt other countries' definitions.

This systematic review (SR) protocol describes an approach which will be used to review the published literature regarding the criteria used to define RDs and ODs from both qualitative and quantitative points of view, and to explore the rationale behind each criterion.

## REVIEW AIMS AND OBJECTIVES

### Aim

This study aims to perform a systematic literature review to identify the criteria used to define RDs and ODs from both qualitative and quantitative perspectives.

### Objectives

1. To identify the quantitative criteria used to define RDs and ODs.
2. To identify the qualitative criteria used to define RDs and ODs.
3. To explore the rationale behind the quantitative criteria used to define RDs and ODs.
4. To explore the rationale behind the qualitative criteria used to define RDs and ODs.
5. To explore the methodological characteristics that explain any heterogeneity in the results.

## METHODS AND ANALYSIS

This protocol is registered in PROSPERO (CRD42021252701) and will follow the PRISMA-P (2015 guidelines).[32]

### Inclusion criteria

The study will include any scientific publication of any design which discusses RDs and/or ODs and answer the research question: what are the criteria to define RDs and ODs globally? Publications will be eligible for inclusion in the SR based on the framework known as the population, intervention, comparator, and outcome (PICO) framework.[33] There will be no restrictions on publication year or jurisdiction (North America, South America, Asia, Europe, Africa, Oceania or the Middle East). We will include studies published in the English language reporting data for the general human population, with data extracted separately for children and adults if reported separately. The comparator element of the PICO framework will not be applicable for this study. Eligible publications will be analysed by searching for general definitions of RD and OD. These definitions will be categorised based on qualitative and quantitative criteria as an outcome of the SR.

### Exclusion criteria

RD due to infection will be excluded as some infections, such as tuberculosis, may be rare in wealthy countries, while being relatively common in poorer countries.[34] Exposure to toxic substances can cause RD, which will also be excluded from this SR. For example, asbestos (fibrous silicate mineral) exposure can cause mesothelioma (a type of cancer that develops in the thin layer of tissue that covers the lungs and chest wall).[34] Furthermore, rare forms of cancer will be excluded in our SR, as there are discrepancies in rarity between countries that are typically captured by national cancer registries; for example, hepatocellular carcinoma is common in China (95.7/100 000 people) but is relatively uncommon in Canada (6.8/100 000 people).[35 36]

### Search strategy

A systematic literature review will be conducted using Boolean Operator rules ('AND', 'OR' and 'NOT') with truncation to broaden the search terms to account for various word spellings and endings. The following search terms were discussed and validated by the research team along with a library information specialist, see online supplemental appendix 1: ultra-orphan disease; ultra-orphan drug; ultra-rare disease; orphan disease; orphan drug; orphan medicinal product; orphan product; orphan subset; orphan indication; orphan pharmaceutical product; orphan designation; neglected disease; rare and neglected disease; rare condition; rare disease; rare disorder; rare disability; rare medicinal technology; syndrome without a name; undiagnosed disease; lifethreatening; debilitating; severe and intractable; highly specialised technology; very rare disease; low-frequency disease; pharmacological therapies of high complexity; priority review drugs; extremely rare disease; and orphan drug reimbursement system.

Publications will be retrieved from various databases, including PubMed, Medline, EMBASE, Scopus, Web of Science (science and social science citation index) and grey literature databases, including GreyLit and Open-Grey, where additional legal terms (rare disease strategy, rare disease policy, rare disorder initiative, orphan disease declaration, rare disease national plan, orphan drug act, rare disease act, orphan drug regulation, orphan medical product decision, orphan drug directive and orphan drug recommendation) will be used in the search strategy. To maximise literature saturation, this search will be complemented by reverse citation screening of relevant publications and forward citation searching. Databases will be searched from their commencement to 17 December 2021.

### Study selection

Studies will be selected after publications are compiled, and duplicates will be removed. Two authors (GMA and KK) will then independently conduct two rounds of screening of the titles and abstracts to assess initial eligibility based on the inclusion and exclusion criteria. The full text of both the publications that appear to meet the inclusion criteria, and those where there is insufficient information in the title and abstract to exclude the study,

will be retrieved. Disagreements between the two individual reviewers will be resolved by a third reviewer (AM) and all decisions will be documented in a Microsoft Excel spreadsheet. None of the reviewers will be blinded to the journal or publication titles, the study or publication authors or the institutions.

## Data extraction

The included articles will be summarised using a custom spreadsheet developed using the primary screening, quality assessment and a data extraction web-based software platform tool, Covidence.[37] The data will be extracted independently by two reviewers (GMA and KK). Where there is missing or unreported data, or if additional details are required, the study's primary investigator (GMA) will attempt to contact the study authors. The timeframe for a reply before the article is excluded based on the lack of requested information will be 3 weeks.

The extracted data will include author(s) name(s), publication year, publication type, journal title, study design, organisation, geographical location, population, results of a study quality assessment and the cited definitions of RD and OD. Moreover, the types of RD and OD criteria (qualitative or quantitative), their descriptor criteria, the rationale behind the definition, the status of the definition (developed or adopted) and whether RD and OD definitions have been considered (implicit or explicit) will also be recorded.

## Critical appraisal of studies

Data extraction will be performed and each study's methodological rigour will be critically appraised by two authors (GMA and KK) separately using customised forms based on the Joanna Briggs Institute (JBI) checklists.[38] The JBI's critical appraisal tools assist in assessing the trustworthiness, relevance and results of published studies. Also, the grey literature will be evaluated and critically appraised using an AACODS checklist to assess its authority, accuracy, coverage, objectivity, date and significance.[39]

## Data synthesis

Narrative synthesis will be undertaken using multiple studies to summarise and explain the current state of knowledge in relation to the systematic review's aim and objectives. An initial description of the results from each included article will be developed using a preliminary synthesis in the form of thematic and content analyses for qualitative descriptors, while descriptive analysis will be conducted in the form of frequencies and percentages for quantitative descriptors, and the results will be presented in a tabular format.

These data will be processed using various quantitative statistical and qualitative analysis approaches to investigate the similarities and differences between the studies, and to explore the relationships within the data, with the aim of reporting key elements for defining RDs and ODs both qualitatively and quantitatively. The risk of bias will be assessed by considering the diversity of the study designs, populations and outcomes.

## Patient and public involvement

During the protocol development, there was no public or patient involvement. As the SR is focused on a secondary analysis of literature in the public domain, there will be no patient or public involvement in the SR.

## Ethics and dissemination

Ethical committee approval is not required to conduct this SR, as no primary data will be collected; this SR is focused on a secondary analysis of the literature in the public domain. It is part of a PhD project at Queen's University Belfast, and the results will be presented at conferences and published as a thesis. We anticipate publication in a peer-reviewed journal. The results will guide stakeholders in defining RDs and ODs, and will inform policy decisions and future research.

**Acknowledgements** The authors would like to thank Richard Fallis, a library information specialist, for his assistance in verifying the search terms and strategy to be used in this study.

**Contributors** AM and HAA-O were responsible for research supervision. GMA, AM and HAA-O conceived, conceptualised and designed this study. GMA, KK, AM and HAA-O will contribute to data acquisition and analysis. All authors contributed to the interpretation of results. GMA drafted this manuscript. All the authors have reviewed the protocol for important intellectual content, approved the final version submitted for publication and agreed to be accountable for all aspects of this research.

**Funding** This systematic review is supported by the Medical Research Council's Northern Ireland Executive in support of the Northern Ireland Genomic Medicine Centre through the Belfast Health and Social Care Trust (award number: MC_PC_16018) and the Science Foundation Ireland and Department for the Economy, Northern Ireland partnership (award number: 15/IA/3152). Princess Nourah University, Riyadh, Saudi Arabia, is supporting this study through a PhD scholarship with no grant number. These funders played no role in the development of this protocol.

**Competing interests** None declared.

**Patient and public involvement** Patients and/or the public were not involved in the design, or conduct, or reporting, or dissemination plans of this research.

**Patient consent for publication** Not applicable.

**Provenance and peer review** Not commissioned; externally peer reviewed.

**ORCID iDs**
Ghada Mohammed Abozaid http://orcid.org/0000-0002-4523-0577
Katie Kerr http://orcid.org/0000-0002-8469-8885
Amy McKnight http://orcid.org/0000-0002-7482-709X
Hussain A Al-Omar http://orcid.org/0000-0002-0765-0466

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
