## [Reviewer comments · BMJ Open]

ARTICLE DETAILS

TITLE (PROVISIONAL)	Criteria to Define Rare Diseases and Orphan Drugs: A Systematic Review Protocol
AUTHORS	Abozaid, Ghada; Kerr, Katie; McKnight, Amy; Al-Omar, Hussain A.

VERSION 1 – REVIEW

REVIEWER	Valdez, R Centers for Disease Control and Prevention, Atlanta
REVIEW RETURNED	19-Apr-2022

GENERAL COMMENTS	The plan for the systematic review seems comprehensive. I would recommend the use of more legal terms in their search, specially for the grey literature search (e.g., strategy, initiative, declaration, national plan, act, regulation, decision, directive, recommendation, etc.). Most rare disease definitions come from legislative bodies. By the way, the definition for the US comes from Congress (Orphan Drug Act of 1983 as amended in 1984) not the FDA. The English needs minor editing and spell checking. The attached file could be useful for the authors.
--

REVIEWER	Giannuzzi, Viviana Fondazione per la Ricerca Farmacologica Gianni Benzi ONLUS
REVIEW RETURNED	03-May-2022

GENERAL COMMENTS	unfortunately the manuscript lack of some important issues. Only the introduction and methods are provided. No results nor discussion nor conclusions are available. The introduction is confusing and lack updated and clear topics related with the aim of the manuscript. Minor comments: Not all rare diseases are chronic. Please amend the abstract. A reference is needed for the sentence "RDs affect approximately 6% of the worldwide population", which equates to >470 million persons". In the introduction, it should be better explained some mentioned definitions in the EU, US and Japan come from the laws on orphan drugs.
--

VERSION 1 – AUTHOR RESPONSE

Reviewer: 1

Dr. R Valdez, Centers for Disease Control and Prevention, Atlanta,

Comments to the Author: The plan for the systematic review seems comprehensive.

1. I would recommend the use of more legal terms in their search, especially for the grey literature search (e.g., strategy, initiative, declaration, national plan, act, regulation, decision, directive, recommendation, etc.).

2. Most rare disease definitions come from legislative bodies. By the way, the definition for the US

comes from Congress (Orphan Drug Act of 1983 as amended in 1984) not the FDA.

3. The English needs minor editing and spell checking. The attached file could be useful for the authors.

Comment 1: [I would recommend the use of more legal terms in their search, especially for the grey literature search (e.g., strategy, initiative, declaration, national plan, act, regulation, decision, directive, recommendation, etc.)]

Response: I agreed with you. Thank you for highlighting this issue. Therefore, we will consider these legal terms as part of our grey literature search. The changes can be found in the revised manuscript with red colour in page number 8, 1st paragraph, and lines 219 -222.

Changes: [where additional legal terms (rare disease strategy, rare disease policy rare disorder initiative, orphan disease declaration, rare disease national plan, orphan drug act, rare disease act, orphan drug regulation, orphan medical product decision, orphan drug directive, orphan drug recommendation) will be used in the search strategy]

Comment 2: [Most rare disease definitions come from legislative bodies. By the way, the definition for the US comes from Congress (Orphan Drug Act of 1983 as amended in 1984) not the FDA.]

Response: We have revised the rare disease definition in United States based on the Orphan Drug Act (ODA) of 1983 and Rare Diseases Act (RDA) of 2002 (legislative bodies); this is incorporated to the revised manuscript on page number 4-5, 2nd paragraph, lines 115 -127. The FDA used the definition of RD for orphan drugs designation status by combining both definitions from ODA and RDA.

Changes: [In the United States (US), the Rare Diseases Act (RDA) of 2002, defines rare disease precisely according to prevalence as "rare disease' means any disease or condition that affects less than 200,000 persons in the United States". Prior to the RDA was the Orphan Drug Act (ODA) of 1983, a federal law which facilitate the development of rare diseases drugs. Define rare diseases based on qualitative descriptor as "the term 'rare disease or condition' means any disease or condition which occurs so infrequently in the United States that there is no reasonable expectation that the cost of developing and making available in the United States a drug for such disease or condition will be recovered from sales in the United States of such drug". While, the Food and Drug Administration (FDA) used both qualitative and quantitative criterion to define RD as "any disease or condition that affects less than 200,000 people in the United States or affects >200,000 in the United States and for which there is no reasonable expectation that the cost of developing and making available in the United States a drug for such disease or condition will be recovered from sales in the United States of such drug"3, 4

Comment 3: [The English needs minor editing and spell checking.]

Response: Thank you for the comment. Spelling errors and grammars have been corrected and certified by a proofreader (Editage®).

Reviewer: 2

Dr. Viviana Giannuzzi, Fondazione per la Ricerca Farmacologica Gianni Benzi ONLUS

Comments to the Author: Dear author,

1. Unfortunately, the manuscript lack of some important issues. Only the introduction and methods are provided. No results nor discussion nor conclusions are available.
2. The introduction is confusing and lack updated and clear topics related with the aim of the manuscript.
3. Minor comments: Not all rare diseases are chronic. Please amend the abstract.
4. A reference is needed for the sentence "RDs affect approximately 6% of the worldwide population', which equates to >470 million persons".
5. In the introduction, it should be better explained some mentioned definitions in the EU, US and Japan come from the laws on orphan drugs.

Comment 1: [unfortunately the manuscript lack of some important issues. Only the introduction and methods are provided. No results nor discussion nor conclusions are available.]

Response: Thank you for providing a review of this manuscript. We would highlight that this manuscript describes a systematic review protocol and therefore no results, discussion or conclusions should be included at this stage. We have followed the detailed instructions to authors for submitting a protocol to BMJ Open "Protocol manuscripts should report planned ... research studies; We encourage the submission of protocol manuscripts at an early stage of the study"⁵. The systematic review protocol is defined based on the Preferred Reporting Items for Systematic reviews and Meta-analyses (PRISMA) guideline⁶ as a document that "describes the rationale, hypothesis, and planned methods of the review. It should be prepared before a review is started and used as a guide to carry out the review." The PRISMA-P 2015 checklist supports the development and transparent reporting of protocols. The BMJ open Journal requires a completed PRISMA-P 2015 checklist as a condition of submission of systematic review protocols, by ensuring the protocol adheres to these basic reporting sections which are administrative information, introduction, and methods,^[5]. Publishing protocols in this manner improves transparency and can help in preventing research duplication.

Comment 2: [The introduction is confusing and lack updated and clear topics related with the aim of the manuscript.]

Response: The aim of this manuscript is to describe a protocol for a systematic review that will identify criteria used to define rare diseases and orphan drugs, providing a rationale and evidence-based for future research. Rare diseases are a major public health concern affecting 1 in every 17 persons at some point in their lives⁷. There is no global consensus definition for rare diseases, which causes challenges in the identification, reporting and treatment of rare diseases. Rare diseases are not explicitly coded in many health care systems, nor Health Research Classification Systems, making it challenging to effectively evaluate activity and outcomes within health research portfolios. Compounding the issue, pharmaceutical companies invest to ensure a return on their investment. They are unwilling to invest with little is known about disease, and relatively their prevalent. This leaves a vast, unmet need of patients and caregivers living with a rare disease, in particular those seeking treatment with orphan drugs.

The introduction contained in this research protocol sets the scene for this review by highlighting the global prevalence of rare diseases, providing exemplars of differing rare disease and orphan drug criteria, and explaining why the review will consider both qualitative and quantitative approaches. Once the review is completed, the introduction to the actual review article will provide further details, but it would be pre-emptive to include data here that should be revealed through the comprehensively described review process.

Comment 3: [Not all rare diseases are chronic. Please amend the abstract.]

Response: Thank you for your comment, we agree with the fact that not all RDs are chronic diseases and we have incorporated your suggestion in abstract in 2nd page, 1st paragraph, and line 37. Although European Union (EU) used "chronically debilitating conditions" as a qualitative descriptor to define RDs, we appreciate that rare diseases may be acute or chronic and have updated the abstract to clarify this important point

Changes: [Rare diseases (RDs) are often chronic and progressive life-threatening medical conditions affecting a low percentage of the population compared to other diseases.].

Comment 4: [A reference is needed for the sentence "RDs affect approximately 6% of the worldwide population", which equates to >470 million persons".]

Response: Thank you so much for catching this confusing issue. We added references to the sentence as requested. This percentage (6%) we used it to calculate the estimate number of population had RDs. We revise the manuscript, and the change can be found with a red colour – page number 3, 3rd paragraph, and lines 71 -72.]

Changes: [RDs affect approximately 6% of the worldwide population⁸, when extrapolated to the worldwide population in 2022 (7 950 418 550 people)⁹ it will be equivalent to >470 million persons]

Comment 5: [In the introduction, it should be better explained some mentioned definitions in the EU, US and Japan come from the laws on orphan drugs.]

Response: Thank you very much for highlighting this important point. We have updated the manuscript to emphasize that some definitions derived from laws on orphan drugs, like Orphan Drug Act (ODA) of 1983 and Rare Diseases Act (RDA) of 2002. Moreover, we adapted the search strategy to include more legal terms for grey literature databases. Presented in manuscript with red colour on page number 4-5,

2nd paragraph, lines 115 -127 and on page number 8, 1st paragraph, and lines 214 -217. When completed, our review will provide a detailed summary of individual definitions of rare diseases and orphan drugs, clearly highlighting their origin.

Changes1: [In the United States (US), the Rare Diseases Act (RDA) of 20021, defines rare disease precisely according to prevalence as "'rare disease' means any disease or condition that affects less than 200,000 persons in the United States". Prior to the RDA was the Orphan Drug Act (ODA) of 19832, a federal law which facilitate the development of rare diseases drugs. Define rare diseases based on qualitative descriptor as "the term 'rare disease or condition' means any disease or condition which occurs so infrequently in the United States that there is no reasonable expectation that the cost of developing and making available in the United States a drug for such disease or condition will be recovered from sales in the United States of such drug"3, 4

Changes2: [where additional legal terms (rare disease strategy, rare disease policy rare disorder initiative, orphan disease declaration, rare disease national plan, orphan drug act, rare disease act, orphan drug regulation, orphan medical product decision, orphan drug directive, orphan drug recommendation) will be used in the search strategy]

Rebuttal References:

1. Health NIo. PUBLIC LAW 107-280—NOV. 6, 2002. In: 2002 RDA, ed., 2002.
2. Health. NIo. Public Law 97-414 97th Congress. In: Health NIo, ed., 1983.
3. US Code. 21 USC 360ee: Grants and contracts for development of drugs for rare diseases and conditions 2022, July 1 [Available from: <https://uscode.house.gov/view.xhtml?req=granuleid:USC-prelim-title21-section360ee&num=0&edition=prelim>.
4. U.S. Food & Drug Administration. DESIGNATION OF DRUGS FOR RARE DISEASES OR CONDITIONS SEC. 526 OF THE FEDERAL FOOD, DRUG, AND COSMETIC ACT [21 USC 360bb]. 2018 [Available from: <https://www.fda.gov/industry/designating-orphan-product-drugs-and-biological-products/orphan-drug-act-relevant-excerpts>.
5. BMJ open. Submission guidelines 2022 [Available from: <https://bmjopen.bmj.com/pages/authors/#protocol>.
6. Moher D, Shamseer L, Clarke M, et al. Preferred reporting items for systematic review and meta-analysis protocols (PRISMA-P) 2015 statement. *Syst Rev* 2015;4:1. doi: 10.1186/2046-4053-4-1 [published Online First: 20150101]
7. team RDp. The UK Strategy for Rare Diseases: Rare Diseases implementation plan for England. In: Care DoHaS, ed., 2018, January 29.
8. Nguengang Wakap S, Lambert DM, Olry A, et al. Estimating cumulative point prevalence of rare diseases: analysis of the Orphanet database. *Eur J Hum Genet* 2020;28(2):165-73. doi: 10.1038/s41431-019-0508-0 [published Online First: 20190916]
9. Population matters. CURRENT WORLD POPULATION 2022 [Available from: https://populationmatters.org/population-numbers?gclid=EAIaIQobChMIjcIIIO5-AIVA853Ch0PnwM_EAAYASAAEgIp7vD_BwE.